# Uric Acid in Inflammation and the Pathogenesis of Atherosclerosis

**DOI:** 10.3390/ijms222212394

**Published:** 2021-11-17

**Authors:** Yoshitaka Kimura, Daisuke Tsukui, Hajime Kono

**Affiliations:** 1Department of Internal Medicine, Faculty of Medicine, Teikyo University of Medicine, Tokyo 173-8605, Japan; yo.kimura.july@med.teikyo-u.ac.jp (Y.K.); tsukui@med.teikyo-u.ac.jp (D.T.); 2Department of Microbiology and Immunology, Faculty of Medicine, Teikyo University of Medicine, Tokyo 173-8605, Japan

**Keywords:** hyperuricemia, atherosclerosis, inflammation, reactive oxygen species

## Abstract

Hyperuricemia is a common metabolic syndrome. Elevated uric acid levels are risk factors for gout, hypertension, and chronic kidney diseases. Furthermore, various epidemiological studies have also demonstrated an association between cardiovascular risks and hyperuricemia. In hyperuricemia, reactive oxygen species (ROS) are produced simultaneously with the formation of uric acid by xanthine oxidases. Intracellular uric acid has also been reported to promote the production of ROS. The ROS and the intracellular uric acid itself regulate several intracellular signaling pathways, and alterations in these pathways may result in the development of atherosclerotic lesions. In this review, we describe the effect of uric acid on various molecular signals and the potential mechanisms of atherosclerosis development in hyperuricemia. Furthermore, we discuss the efficacy of treatments for hyperuricemia to protect against the development of atherosclerosis.

## 1. Introduction

Recently, the prevalence of hyperuricemia has been increasing among developed or high-income developing countries [1]. According to the National Health and Nutrition Examination Survey (NHANES) 2007–2016, the prevalence of hyperuricemia in the U.S. was 19.1% in 1988–1994, but it increased to 21.5% in 2007–2008 [2]. Since then, the prevalence has remained at this relatively high level. According to the report for 2015–2016, the prevalence was 20.2% in men and 20.0% in women [3].

Since ancient times, gout has been known to be associated with rich foods and excessive alcohol consumption. For this reason, gout has been referred to as the “disease of kings”. In some eras, gout was perceived as socially desirable because of its prevalence among the politically and socially powerful [4]. The prevalence differs between regions because of variations in lifestyle [2]. Gout is associated with Western diets or lifestyles, so the spread of Western diets in non-Western countries led to an increased prevalence of gout. A Portuguese missionary who visited Japan in the 16th century reported that the country had few gout patients, unlike in Europe. The first report of a patient with gout in Japan was in 1898 [5]. Patients with gout were rare until the end of World War II, but after the 1960s, its prevalence rapidly increased as people adopted the Western lifestyle. At present, serum uric acid levels remain low in regions without a predominantly Western lifestyle [2].

Hyperuricemia is closely associated with hypertension [6], chronic kidney diseases [7], type 2 diabetes mellitus [8], and metabolic syndrome [9], which are well known to be related to atherosclerosis. The Western diet is also closely related to the progression of atherosclerosis. For the last half-century, the association between hyperuricemia and coronary artery diseases has been discussed in many epidemiological studies. Several studies have reported a positive correlation between hyperuricemia and cardiovascular diseases [10,11], while others have found no association [12,13]. However, several meta-analyses of prospective studies indicating hyperuricemia’s association with cardiovascular diseases have been reported recently (see below).

Recently, chronic inflammation and the inflammatory signaling pathway have been shown to play essential roles in the development of atherosclerosis. In this review, we focus on the effect of uric acid on the inflammatory pathway and show the role of uric acid in the pathogenesis of atherosclerosis.

## 2. Mechanisms of Hyperuricemia

Uric acid is an end product of purine metabolism in humans. Purine is metabolized to hypoxanthine, and then xanthine dehydrogenase (XDH) oxidizes hypoxanthine to xanthine and then xanthine to uric acid. Uric acid is metabolized to allantoin by uricase (uric acid oxidase), which is expressed in the liver in mammals, except for Hominidae primates, such as humans, chimpanzees, gorillas, and orangutans. Serum uric acid levels in most mammals are extremely low, around 0.5–1 mg/dL, because of the presence of uricase [14]. Primates such as humans that have lost uricase have several mechanisms to regulate serum uric acid levels. One is the increased excretion of uric acid by uric acid transporters [15], and the other is the increased activity of hypoxanthine-guanine phosphoribosyl transferase (HGPRT), which recycles purines [16]. Hence, these effects limit the elevation of serum uric acid levels to only about 1–2 mg/dL.

Two-thirds of the excretion of uric acid is from the kidney, and one-third is from the intestinal tract. Several urate transporters have been identified. Among them, NPT1, NPT4, OAT1, OAT2, OAT3, OAT4, URAT1, GLUT9, and ABCG2 are closely related to serum uric acid levels. NPT1, NPT4, OAT1, OAT2, OAT3, and ABCG2 are related to urate secretion, and OAT4, URAT1, and GLUT9 are associated with urate reabsorption [17,18]. These transporters are expressed in the kidney, while ABCG2 is also expressed in the intestinal tract. One of the causes of hyperuricemia is the mutation of HGPRT, which converts hypoxanthine to inosine monophosphate (IMP) and guanosine to guanosine monophosphate (GMP). The decreased activity of HGPRT results in increased hypoxanthine levels and hyperuricemia. HGPRT deficiency is known to cause Lesh–Nyhan syndrome or Kelly–Seegmiller syndrome [19].

Other than genetic factors, the risk factors of hyperuricemia and gout are as follows: the intake of purine-rich foods such as meat or seafood; the intake of beverages and food that contain high amounts of sugar, especially fructose; and the intake of alcohol, especially beer. Consuming these items causes the intake of excessive amounts of purine. Fructose is phosphorylated to fructose-1-phosphate by using ATP in the glycolysis cascade. Excess intake of fructose results in the consumption of large amounts of ATP, the production of ADP and AMP, and the enhancement of AMP deaminase activity, which converts AMP to IMP. This accelerates the production of uric acid [19,20]. Hypertension, metabolic syndrome, chronic kidney disease, high body mass index (BMI), and obesity are also known risk factors of hyperuricemia. One of the mechanisms of hyperuricemia is insulin resistance, which affects the renal clearance of uric acid [21] (Figure 1).

## 3. Epidemiology of Association between Uric Acid and Atherosclerosis

The association between serum uric acid level and cardiovascular risk has been studied for several decades. Several studies have suggested that serum uric acid (SUA) is correlated with cardiovascular disease, but some studies have reported contradictory results. However, recent meta-analyses of prospective studies have supported that hyperuricemia is an independent risk factor of cardiovascular diseases.

Zuo et al. performed a meta-analysis of 14 prospective studies comprising 341,389 adult participants. They reported that the relative risk (RR) of mortality from coronary heart diseases (CHD) was increased in patients with hyperuricemia (RR: 1.14, 95% confidence interval [CI]: 1.06–1.23). Notably, the overall risks of CHD and all-cause mortality were increased by 20% and 9%, respectively, for every 1 mg/dL increase in serum uric acid. According to gender subgroup analyses, hyperuricemia increased the risk of CHD mortality to a greater extent in women compared with men [22]. In addition, Li et al. reported that a meta-analysis of 29 prospective cohort studies (N = 958,410) showed that hyperuricemia was associated with increased risk of CHD morbidity (adjusted RR 1.13; 95% CI 1.05–1.21) and mortality (adjusted RR 1.27; 95% CI 1.16–1.39). In this report, the CHD risk was also greater in women; the RR of CHD mortality for every 1 mg/dL increase in uric acid was 1.02 in males and 2.44 in females [23]. Another meta-analysis that included 11 prospective studies with 172,123 patients suggested that elevated SUA increased the risk of all-cause mortality (RR 1.24; 95% CI 1.09–1.42) and cardiovascular mortality (RR 1.37; 95% CI 1.19–1.57). However, in this study, the risk of all-cause mortality was associated with SUA in males but not in females, unlike in the previously mentioned studies [24].

In Mendelian randomization, the measured variation in genes of a known function is used to examine the causal effect of modifiable exposure to a disease; this method is used in observational studies. Several studies have detected a causal relationship between SUA level and CHD by using Mendelian randomization. Chiang et al. reported that hyperuricemia was associated with an increased cumulative lifetime risk of cardiovascular disease [25]. In another study, every 1 mg/dL increase in genetically predicted uric acid concentration had a significant effect on cardiovascular death (Hazard Ratio [HR], 1.77; 95% CI. 1.12–2.81) and sudden cardiac death (HR, 2.41; 95% CI, 1.65–5.00) [26]. However, other studies reported no significant relationship between hyperuricemia and CHD [27]. 

A recent study indicated that in addition to high SUA levels, variable uric acid levels might be a risk factor for cardiovascular diseases. A gout attack is known to occur when the SUA level fluctuates. The risks of major adverse cardiovascular events (MACE), myocardial infarction (MI), cardiovascular death, heart failure-related hospitalization, and total cardiovascular events were significantly higher in patients with highly variable SUA levels compared with those with stable SUA levels [28]. Fluctuating uric acid levels, such as those in gout attacks, may induce inflammation in coronary arteries and lead to the development of atherosclerosis.

## 4. The Role of Hyperuricemia in the Pathogenesis of Atherosclerosis

As mentioned above, many epidemiological studies have shown an association between hyperuricemia and atherosclerosis-related events. In addition, basic studies have characterized the role of uric acid in the pathogenesis of atherosclerosis.

Atherosclerotic plaques develop as follows. First, vascular endothelial injury caused by mechanical stress induces the expression of adhesion factors, and chemokines and monocytes attach to vascular endothelial cells and infiltrate the subendothelial layer. In the subendothelial layer, monocytes differentiate into macrophages. A high level of hyper-low-density lipoprotein (LDL) cholesterolemia results in the accumulation of LDL in the subendothelial layer. Reactive oxygen species (ROS) oxidize LDL to oxidized LDL (oxLDL), which induces inflammation in vascular endothelial cells. The recruited monocytes/macrophages take in oxLDL and differentiate into foam cells. Foam cells secrete inflammatory cytokines and various chemokines, which attract leukocytes such as T cells, other lymphocytes, and dendritic cells. These leukocytes produce TGF-β, which attracts fibroblasts and promotes the formation of fibrous caps, leading to the development of the atheromatous plaque. Oxidative stress and various intracellular signaling cascades are involved in this process, and uric acid affects them in a pro-atherogenic manner.

### 4.1. Oxidative Stress

Oxidative stress is one of the most critical factors in the development of atherosclerosis. Oxidative stress contributes to the pathogenesis of atherosclerosis via induction of the dysfunction of endothelial cells and vasodilation, induction of inflammation in inflammatory cells such as macrophages, aggregation of platelets, and oxidation of LDL.

Reactive oxygen species (ROS) are derived from oxygen molecules (O_2_) and are unstable and powerful oxidizing agents. In vivo, they are produced in the process of oxidative phosphorylation (OXPHOS) in mitochondria. In addition, NADPH oxidases, xanthine oxidases, and lipoxygenases are known to produce ROS. ROS have essential physiological roles in preventing infection and mediating signal transduction. ROS oxidize intracellular proteins, lipids, and DNA and lead to cellular damage. There are inherent systems for antioxidant defense in the body. However, if this balance is lost and oxidative stress becomes dominant, atherosclerotic lesions progress. The level of oxidative stress is correlated with age [29], blood pressure [30], smoking [29], LDL cholesterol [31], and blood sugar level [32]. Patients with CHD have high levels of oxidative stress. 

Uric acid itself is chemically characterized as an antioxidant. In particular, uric acid is an important antioxidant in the extracellular space. Most mammals can synthesize ascorbic acid, a powerful antioxidant, but some primates, including humans, do not have the enzyme required for ascorbic acid synthesis. Thus, uric acid is thought to work as an antioxidant instead of ascorbic acid [33]. In a prior study, uric acid suppressed ROS accumulation and protected against ischemic neuronal injury [34]. It is also known that exercise-induced acute kidney injury can occur in patients with hereditary renal hypouricemia. In such cases, kidney injury results from the insufficient removal of exercise-induced oxidative stress due to the deficiency of uric acid as an antioxidant [35].

However, it is also known that intracellular uric acid plays a role in inducing oxidative stress. In the pathogenesis of atherosclerosis, hyperuricemia acts as an inducer of oxidative stress. The mechanisms by which oxidative stress accumulates under hyperuricemic conditions are as follows:
ROS are produced due to the increased activity of xanthine oxidase in the metabolic process of uric acid;The expression and activity of NADPH oxidase increase;Mitochondrial ROS (mtROS) are produced due to mitochondrial injury.

#### 4.1.1. Xanthine Oxidoreductase

Xanthine oxidoreductase (XOR) exists in two forms, which are xanthine dehydrogenase (XDH) and xanthine oxidase (XO). XDH oxidizes substrates with NAD+ and produces NADH, while XO oxidizes substrates with O_2_ and produces O_2_- or H_2_O_2_. Only mammals have the XO type [36]. XDH is expressed in the liver and small intestine and released to the plasma, where it is converted into XO by a protease [37]. Most XOR exists as the XO type in vascular endothelial cells. XO plays a role in preventing infections by producing ROS in vascular endothelial cells in physiological conditions.

The accumulation of XO in atherosclerotic plaque leads to the production of ROS. An elevated ratio of XO to XDH was observed in response to oscillatory shear stress in plaques [38]. Furthermore, the expression of XO itself was increased in plaques [39]. ROS derived from XO is involved in a vascular endothelial injury. It has been reported that XOR inhibitors improved endothelial dysfunction in patients with chronic heart disease, diabetes mellitus with mild hypertension, smoking, and sleep apnea syndrome (SAS) [40,41,42,43]. The activity of XO is elevated in pathological conditions such as MI and ischemia–reperfusion [44]. Increased XO activity results in a burst of ROS, the attraction of neutrophils, and the induction of tissue injury [45]. 

The production of ROS by XO induces the migration, proliferation, and production of monocyte chemotactic protein-1 (MCP-1) in arteriolar smooth muscle cells [46] and contributes to the development of atherosclerosis. XOR contributes to foam cell formation. Knock-down of XOR suppressed lipid intake in macrophages and their differentiation into foam cells [47]. XOR was also reported to regulate lipid accumulation and be involved in adipocyte differentiation via activation of the transcription factor PPARγ [48]. XOR regulates inflammatory cytokine secretion. Increased plasma XOR activity was correlated with plasma IL-6 level and NF-kB activity [49]. In mouse macrophages, XOR regulated IL-1β secretion via NLRP3 inflammasome activation [50].

#### 4.1.2. NADPH Oxidase

NADPH oxidase (NOX) is a complex of membrane-bound enzymes. In phagocytes, ROS produced by NADPH oxidase eliminates microorganisms. Four members of the NOX family (NOX1, NOX2, NOX4, NOX5) are expressed in vascular smooth muscle cells, endothelial cells, fibroblasts, and perivascular adipocytes. NOX-dependent ROS formation in endothelial cells and vascular smooth muscle cells (VSMCs) potentially activates the expression of adhesion molecules and the subsequent monocyte/macrophage infiltration into the arterial wall. It also contributes to endothelium activation and stimulates VSMC proliferation. NOX1 also contributes to the accumulation of the extracellular matrix and leads to aortic media hypertrophy. Thus, activation of NOX is involved in the pro-atherogenic process [51].

It has been reported that uric acid activates NADPH oxidase and produces ROS. Uric acid-activated NADPH oxidase and increased oxidative stress lead to the activation of p38 MAPK and ERK1/2, causing a subsequent decrease in NO bioavailability and increase in protein nitrosylation and lipid oxidation in adipocytes [52]. In the human aorta, uric acid-treated smooth muscle cells increased the cell proliferation and expression of endothelin-1 (ET-1). These effects were suppressed by a NOX inhibitor and siRNA of the NOX subunit (p47phox). ET-1 is a pro-atherogenic molecule derived from vascular endothelium and promotes strong vasoconstriction, the proliferation of smooth vascular cells, and the proliferation of fibroblasts. The uric acid-induced expression of ET1 is caused by the activation of ERK-AP1 via the production of ROS [53]. In kidney tubular cells, uric acid increased oxidative stress and apoptosis via up-regulation of NOX4 expression and mitochondrial injury. Furthermore, NOX signaling was dependent on the intracellular uptake of uric acid via URAT1, a uric acid transporter [54]. This indicates that uric acid works intracellularly. Uric acid regulates not only the expression level but also the activity level of NOX. Uric acid was observed to promote the phosphorylation of p47phox and the interaction of p-p47phox with p22phox, leading to the assembly of subunits and activation of NOX. In a hepatocyte cell line, uric acid-induced NOX activation caused endoplasmic reticulum (ER) stress, followed by mitochondrial dysfunction and the accumulation of lipids [55]. These results suggest that uric acid is involved in the pathogenesis of the metabolic syndrome.

#### 4.1.3. Mitochondrial ROS

Mitochondrial dysfunction induces cell senescence and apoptosis of vascular endothelial cells and influences the development of atherosclerosis. Decreased copy numbers of mitochondrial DNA (mtDNA), a reduced oxygen consumption rate, and mitochondrial dysfunction were observed in human plaque. Mitochondrial dysfunction was found in the plaque of ApoE-knockout mice fed a high-fat diet. This was attributed to the decreased expression of complexes I, III, IV, and V in OXPHOS, which leads to the disorder of VMSC proliferation and macrophage apoptosis, subsequently increasing the vulnerability to plaques [56,57]. Mitochondrial dysfunction leads to ROS production, thereby increasing the production of inflammatory cytokines and promoting the differentiation of M1 macrophages [58].

Several studies have reported that uric acid induces mitochondrial disorder. A high concentration of uric acid caused increased mitochondrial ROS and mitochondrial damage. In hepatocytes, decreased membrane potential, mitochondrial DNA damage, and suppression of OXPHOS due to decreased levels of cytochrome C and succinate dehydrogenase (SDH) were observed [59]. Soluble uric acid-induced endothelial dysfunction in human aortic endothelial cells was related to reduced mitochondria and ATP production. Mitochondrial DNA damage and accumulation of oxidative stress were increased in hyperuricemic rats [60].

The mechanism of mitochondrial injury induced by uric acid remains unclear, but it is assumed that it involves the production of ROS by NADPH oxidase [54], suppression of AMPK [61], or activation of Rho kinase [62].

### 4.2. Inflammatory Signaling Pathway

As mentioned above, uric acid induces ROS production. ROS are vital mediators that activate various signaling pathways. Furthermore, uric acid itself may activate several intracellular signaling pathways that result in the production of inflammatory cytokines, adhesion factors, and chemokines and regulate cell proliferation and apoptosis, consequently leading to atherosclerosis development (Figure 2). Next, we discuss the role of the signaling pathways activated by uric acid in the pathogenesis of atherosclerosis.

#### 4.2.1. ERK/p38 MAPK Cascade

The intracellular mitogen-activated protein kinase (MAPK) cascade is crucial for bridging extracellular stimuli to intracellular reactions. In atherosclerosis, MAPK transduces the stimulation of cytokines or growth factors to intracellular signals and regulates the growth and survival of cardiomyocytes, smooth muscle cells, or macrophages. The MAPK pathway is composed of three steps of kinase activation: MAPK kinase kinase, MAPK kinase, and terminal MAPK. The main terminal MAPKs are extracellular signal-regulated kinase (ERK) 1/2, Jun N-terminal kinase (JNK), p38 MAPK, and ERK5. ERK, JNK, and p38 have crucial roles in the pathogenesis of atherosclerosis [63].

Activation of p38 MAPK has been characterized as pro-atherogenic. The p38 MAPK promotes the expression of adhesion factors, such as E-selectin and vascular cell adhesion molecule-1 (VCAM-1), and chemokines, such as MCP-1. Furthermore, activation of p38 MAPK and ERK1/2 leads to the proliferation and hypertrophy of VSMCs and the induction of RUNX2, which result in the calcification of the aortic wall and aortic valves. p38 MAPK is associated with the migration and proliferation of VSMCs and is involved in the induction of angiogenesis and the formation of atheromatic plaque. Activation of p38 MAPK promotes the adhesion ability of immune cells via the expression of CXCR2 and the expression of inflammatory cytokines [64]. p38 MAPK is involved in the uptake of LDL [64]. Activation of JNK or p38 MAPK is required for foam cell formation induced by oxLDL [63].

Several reports have shown that uric acid activates p38 MAPK and ERK. ROS were produced in cardiomyocytes exposed to HUA, and ERK and p38 MAPK were sequentially activated. As a result, the viability of the cardiomyocytes exposed to HUA decreased. In vivo, ERK/p38 MAPK was activated in the heart of a high-uric-acid mouse model, indicating that uric acid induces myocardial damage [65,66]. Activation of ERK1/2 and p38 MAPK was also observed in VSMCs and promoted the expression of MCP-1. This activation of MAPK was also caused by the production of ROS by HUA [67]. In pancreatic β-cells, uric acid activated ERK, decreased cell viability, and induced apoptosis and ROS production. Zurampic, a URAT1 inhibitor, inhibited the ERK pathway and attenuated uric acid-induced cell damage [68]. This observation reflects the effect of intracellular uric acid on MAPK activity. 

Uric acid also regulates MAPK via phosphatase activity that inhibits the MAPK pathway. In macrophages, febuxostat activated MAPK phosphatase-1 (MKP-1) and deactivated JNK, which led to the suppression of MCP-1 expression [69].

#### 4.2.2. AMPK

AMP-activated protein kinase (AMPK) is a serine/threonine kinase that regulates the intracellular energy state. AMPK is activated by decreased intracellular ATP concentrations and an increased ratio of ATP to ADP or AMP. The activation of AMPK promotes glycolysis and OXPHOS and leads to the production of ATP. Currently, research is focused on AMPK as a key molecule connecting metabolism with inflammation. The suppression of AMPK induced inflammatory responses, such as the production of inflammatory cytokines in macrophages and activation of the NLRP3 inflammasome [70,71]. In a study on the pathogenesis of atherosclerosis, activation of AMPK suppressed the development of atherosclerosis in ApoE-KO mice [72,73]. Furthermore, atherosclerotic lesions were increased in endothelial cell-specific AMPK-KO mice [74]. The activation of AMPK causes attenuation of the migration of monocytes [72] and anti-inflammatory effects via inhibition of STAT3 and suppression of differentiation of monocytes to macrophages [73]. 

Uric acid has been reported to suppress AMPK. In a fructose-treated hepatocyte cell line, uric acid suppressed AMPK activity and was involved in gluconeogenesis and insulin resistance [75,76]. This suggests that uric acid is involved in the pathogenesis of metabolic syndrome via the regulation of AMPK. However, several studies reported that AMPK was activated by ROS induced by uric acid [77,78].

In a study of atherosclerosis, it was reported that AMPK was activated in blood cells and plaques, and serum IL-1β or TNFα was decreased in a urate-lowering mouse model fed HFD in which uric acid levels were decreased by the administration of allopurinol or overexpression of uricase. In vitro, uric acid attenuated AMPK activity and led to the activation of the NLRP3 inflammasome and the production of IL-1β [61]. Another study also reported the effect of allopurinol on AMPK. AMPK activity was reduced in rats fed a high-fructose diet, but administration of allopurinol rescued the activation of AMPK [79].

#### 4.2.3. PI3K-Akt Pathway

Phosphatidylinositol-3 kinase (PI3K) is a phosphatidylinositol kinase that catalyzes phosphorylation at the 3 position of the inositol ring of phosphatidylinositol, a component of the cell membrane, and also has protein serine/threonine kinase activity. Downstream of PI3K, the most important effector molecule is Akt. Akt is a serine/threonine kinase and a key molecule involved in various signaling pathways, including cell proliferation, differentiation, apoptosis, and migration. Additionally, Akt is involved in the cell cycle and glucose metabolism via GSK3b and cell growth and survival via mTORC1 [80]. With respect to atherosclerosis, the PI3K-Akt pathway regulates the migration of monocytes and macrophages, lipid accumulation, cell proliferation, and endothelial dysfunction, which lead to the development of atherosclerotic plaques [80]. In vitro studies suggested that Akt may play a pro-atherogenic role. However, the development of atherosclerosis was aggravated in Akt1 knockout mice [81]. The role of Akt in atherosclerosis is still debated.

In human monocytes, uric acid was observed to phosphorylate Akt, activate mTOR, and subsequently suppress autophagy. These events resulted in the suppressed expression of IL-1R antagonist and increased production of IL-1β [82]. However, uric acid was also reported to suppress Akt. Uric acid was suggested to be involved in the progression of atherosclerosis via insulin resistance induced by the suppression of Akt [83].

#### 4.2.4. Inflammasome

The inflammasome is an innate immune sensor and regulates the activity of caspase-1. Activation of the inflammasome is induced after the recognition of pathogen-associated molecular patterns (PAMPs) derived from microorganisms and damage-associated molecular patterns (DAMPs) from dead or dying host cells. The nucleotide-binding domain and leucine-rich repeat protein-3 (NLRP3) inflammasome is involved in various infections and inflammatory diseases. The expression of NLRP3 is induced by the activation of NF-kB. Next, it assembles and forms a complex with an adaptor protein, ASC, and procaspase-1. Subsequently, procaspase-1 undergoes autolysis and matures to caspase-1. Caspase-1 processes pro-IL-1β and pro-IL-18 to mature IL-1β and IL-18. At the same time, pyroptosis is induced, and IL-1β is released to the extracellular space [84,85].

Sterile crystals activate the NLRP3 inflammasome. In gout, monosodium urate (MSU) crystals activate the NLRP3 inflammasome, induce the release of IL-1β and promote the development of arthritis. The mechanism of NLRP3 inflammasome activation by MSU crystals involves phagocytes such as macrophages or neutrophils; these cells take up the crystals, and the crystals lead to lysosomal rupture and release of cathepsin B to the cytosol. K+ efflux or production of ROS is also induced and triggers the activation of the NLRP3 inflammasome [84,86]. Recently, MSU crystals were shown to induce the translocation of Nrf2 into nuclei and alter intracellular ROS levels, which promotes activation of the NLRP3 inflammasome [87].

Recently, it has become clear that the NLRP3 inflammasome plays an important role in the pathogenesis of atherosclerosis. Canakinumab, an IL-1β inhibitor, suppressed the development of atherosclerosis [88]. Furthermore, colchicine, which inhibits the formation of the NLRP3 inflammasome, also protects against the recurrence of cardiovascular diseases [89].

In atherosclerotic plaques with hyperuricemia, deposition of MSU crystals was reported [90,91]. MSU crystals can be distinguished from calcium crystals by using dual-energy CT, which is useful for the detection of MSU crystals in gout and urolithiasis. Dual-energy CT was performed on 59 patients with gout and 47 controls, and the frequency of cardiovascular MSU deposition was analyzed. The frequency of the deposition in cardiovascular systems was higher among patients with gout (51 [86.4%]) compared with controls (7 [14.9%]), and in coronary arteries, it was higher among patients with gout (19 [32.2%]) compared with controls (2 [4.3%]) [90]. The expression of XO was increased, and there were significantly higher concentrations of uric acid in atherosclerotic plaques [39], which may also affect the deposition of MSU crystals. However, the findings of dual-energy CT may include artifacts [92]. The association between the deposition of MSU crystals and cardiovascular events is not yet clear. Andres et al. reported that MSU deposition in the knee or first metatarsophalangeal joints was related to calcification of coronary arteries [93].

Soluble uric acid, as well as MSU crystals, has been reported to activate the NLRP3 inflammasome [94]. Soluble uric acid induces the production of mitochondrial ROS and leads to the activation of NLRP3 inflammasome complexes. In another report, soluble uric acid suppressed AMPK, led to the production of mitochondrial ROS, and finally activated the NLRP inflammasome [61].

Inflammasome activation in patients with gout or hyperuricemia has been observed in several studies. Serum IL-18, an inflammasome-related cytokine, was reported to be higher in gout patients, and the serum IL-18 level was correlated with the level of C-reactive protein (CRP) and the erythrocyte sedimentation rate (ESR) [95]. The expression of IL-1β and IL-18 was increased in the peripheral blood mononuclear cells of patients with active gout [96]. A correlation of the plasma uric acid level with the plasma IL-18 level was also reported [97]. Furthermore, decreasing plasma uric acid levels by administering benzbromarone resulted in decreased plasma IL-18 levels [61].

#### 4.2.5. NO, HMGB1, RAA, ER Stress, and Mechanism of Sensing UA

NO regulates blood vessel tonus and acts as an internal anti-atherogenic factor. However, NO is an unstable molecule and easily oxidized by ROS, and subsequently, the physiological activity is lost. A decrease in anti-atherogenic factors is one of the pro-atherogenic effects of ROS. The reactivity of O_2_- itself is weak, but it reacts with NO to produce peroxynitrite (ONOO-), a very powerful radical. In fact, the vasodilation ability was decreased in patients with CHD in whom NADPH oxidase and XOR were activated. Vasodilation impairment in hyperuricemia was improved by the administration of allopurinol, a xanthine oxidase inhibitor. 

As mentioned above, NO is consumed by excessive ROS production, but NO production is also decreased by uric acid. The dephosphorylation of eNOS (endothelial NO synthase) was induced via uric acid transporter, and the production of NO was decreased in human umbilical vein endothelial cells (HUVECs). The decreasing production of NO was attenuated by benzbromarone, a URAT1 inhibitor [98]. Production of eNOS is also regulated by the HMGB1-RAGE pathway. In HUVECs, uric acid promoted the production of HMGB1 and the expression of RAGE, a receptor of HMGB1, and suppressed the expression of eNOS. Furthermore, the autocrine effect of HMGB1 leads to further production of HMGB1. The blocking of RAGE prevented the up-regulation of HMGB1 and the decrease in eNOS expression [99]. In a study to explore the relation between uric acid and HMGB1, uric acid activated the MEK-ERK pathway and regulated intracellular calcium morbidity and HMGB1 acetylation and translocation. The subsequent release of HMGB1 to the extracellular space was promoted, and inflammation was aggravated [100].

The RAA (Renin–Angiotensin–Aldosterone) system may be involved in the effects of uric acid. In HUVECs, uric acid induced the inflammatory response via the up-regulation of renin receptors [101]. Uric acid also up-regulated the expression of angiotensinogen, angiotensin-converting enzyme, and angiotensin II receptors and increased angiotensin II levels. This led to the production of ROS, cell apoptosis, and cellular senescence. The effect was diminished by probenecid, a urate transporter inhibitor [102].

Uric acid was also shown to induce endoplasmic reticulum (ER) stress. In rats with hyperuricemia induced by a uricase inhibitor, oxonic acid, cellular apoptosis, intestinal fibrosis, and diastolic heart dysfunction were exacerbated. Increased expression of calpain-1, a kind of protease, and increased ER stress markers (CHOP, GRP78, p-PERK) were also observed in rat myocardium. In the H9c2 cardiomyocyte cell line, increased apoptosis and ER stress were dependent on calpain-1 [103].

As stated above, intracellular uric acid alters various signaling pathways, including the production of ROS. However, it remains unclear how intracellular uric acid is recognized and activates signaling pathways. Recently, NAIP1 was reported to directly recognize intracellular uric acid and induce the activation of NLRP3 in mice [104]. However, the binding of uric acid with human NAIP1 is weak [104], so the effect of uric acid may be caused by other molecules in humans.

In summary, uric acid plays a pro-atherogenic role in several steps in the progression of plaques as follows. Uric acid promotes oxidative stress and destabilization of NO, which leads to vasoconstriction and endothelial dysfunction. The expressions of chemokines, such as MCP-1, are increased, and monocytes are recruited into the subendothelial layer. Macrophages in subendothelial are differentiated into foam cells depending on oxidative stress by uric acid and the effect of xanthine oxidase. These foam cells or macrophages secrete inflammatory cytokines, and uric acid promotes the production of the cytokines. The inflammatory cytokines attract further inflammatory cells and bring the formation of the necrotic core. Uric acid promotes proliferation and migration of VSMCs via activation of MAPK and oxidative stress, which leads to the progression of atheromatous plaque. Oxidative stress derived from mitochondrial dysfunction by uric acid results in the destabilization of plaques. Inflammation augmented by uric acid via activation of inflammasomes or several inflammatory signaling pathways contributes to the development of atherosclerosis in each atherogenic step (Figure 3).

## 5. The Effect of Therapeutic Agents for Gout on Atherosclerosis

The Guideline for Management of Gout by the American College of Rheumatology (ACR) in 2020 provides the following recommendations [105]. Initiating urate-lowering therapy (ULT) is strongly recommended for gout patients with any of the following: ≥1 subcutaneous tophus; evidence of radiographical damage attributable to gout; or frequent gout flares, with frequent being defined as ≥2 annually. Initiating ULT is not recommended in patients with asymptomatic hyperuricemia. In randomized control trials (RCTs), ULT significantly suppressed gout flares in patients with asymptomatic hyperuricemia. However, the incidence of gout was low (<5%), so the effect of preventing an attack was too costly even in patients with comorbid chronic kidney disease, cardiovascular disease, urolithiasis, and hypertension. In the guideline for gout in Japan, initiating ULT to treat asymptomatic hyperuricemia is recommended in patients with an SUA of more than 8 mg/dL and complications such as chronic kidney disease, urolithiasis, hypertension, cardiovascular disease, diabetes mellites, and metabolic syndrome or in patients whose SUA is more than 9 mg/dL [106]. Thus, the management approach for asymptomatic hyperuricemia varies by region.

Colchicine, nonsteroidal anti-inflammatory drugs (NSAIDs), and steroids are used to treat gout attacks. If they are not effective, the use of an IL-1 inhibitor is considered. When initiating ULT, administrating prophylactic anti-inflammatory therapy such as colchicine and NSAIDs for 3–6 months is recommended [105].

The reported effects of urate-lowering drugs on cardiovascular disease are described below (Table 1). Several small-scale randomized studies have reported the effect of allopurinol. Allopurinol has been the most frequently used xanthine oxidase inhibitor. In a prospective randomized trial of patients with chronic kidney disease (CKD) (eGFR < 60 mL/min/m^2^), administration of allopurinol for 24 months decreased CRP and slowed the progression of CKD. Furthermore, allopurinol reduced the risk of cardiovascular events in 71% of participants [107]. However, in a recent RCT of 369 patients with stage 3 or 4 CKD, allopurinol did not slow the decline in eGFR compared with the placebo [108]. In a meta-analysis of nine RCTs on the efficacy of allopurinol in patients undergoing coronary artery bypass graft (CABG) after ACS or heart failure, the pooled odds ratio of periprocedural ACS and cardiovascular mortality was significantly lower in patients administrated allopurinol during CABG. Nevertheless, the efficacy of allopurinol in the secondary prevention of ACS or mortality in long-term outcomes was not observed [109]. In another randomized controlled prospective trial in 100 patients with ACS, the allopurinol treatment was compared with the placebo. One month of treatment with allopurinol improved the indicators of oxidative stress and inflammatory response (malondialdehyde-modified LDL, oxLDL, high-sensitivity CRP, and TNFα) and significantly increased the level of NO. The symptoms and frequency of angina pectoris were also significantly improved by allopurinol. However, the rates of stent implantation and cardiovascular events during the 2-year follow-up were not significantly different between the allopurinol and control groups [110]. Several large-scale case–control studies showed that allopurinol decreased the risk of cardiovascular diseases such as acute myocardial infarction [111,112]. Another frequently used xanthine oxidase inhibitor is febuxostat. Allopurinol is a purine base analog, while febuxostat is a non-purine analog inhibitor, for which adverse effects related to purine metabolism, such as cytopenia, are less frequent. The effect of febuxostat on renal and vascular function in patients with type 1 diabetes after 8-week administration was examined to explore its endothelial function. Febuxostat had a modest systolic blood pressure-lowering effect, but flow-mediated dilation (FMD) and production of NO were not altered [113]. In a small-scale study comparing febuxostat and benzbromarone, vascular endothelial function was not improved by febuxostat but was improved by benzbromarone [114]. In Japanese patients with asymptomatic hyperuricemia, the development of atherosclerosis was assessed by carotid intima-media thickness. In this RCT, 24 months of febuxostat treatment did not effectively delay the progression of carotid atherosclerosis [115]. In 2018, a large-scale randomized study to compare cardiovascular outcomes associated with allopurinol and febuxostat in patients with gout and cardiovascular disease was reported (CARES study). The study included 6190 patients. The rates of adverse cardiovascular events were not significantly different between febuxostat and allopurinol. However, all-cause mortality and cardiovascular mortality were higher with febuxostat than with allopurinol (HR for death from any cause, 1.22 [95% CI, 1.01–1.47]; HR for cardiovascular death, 1.34 [95% CI, 1.03–1.73]). However, a limitation of this study was its high drop-out rate. The trial regimen was discontinued in 56.6% of patients, and 45.0% discontinued follow-up [116]. After the CARES study, in 2020, a large-scale RCT was reported on the cardiovascular safety of febuxostat (FAST study). Patients who were 60 years or older, already receiving allopurinol, and had at least one additional cardiovascular risk factor were eligible, and 6128 patients were enrolled. Febuxostat was not associated with an increased risk of cardiovascular events, death, or serious adverse events compared with allopurinol [117].

As a uricosuric agent, probenecid was associated with a decreased risk of MI, stroke, and chronic heart failure exacerbation and mortality compared with allopurinol in patients with gout who were 65 years or older in a retrospective, propensity score-matching study [118].

In several studies described above, ULT was effective for the prevention of cardiovascular events. However, a recent meta-analysis of RCTs in which cardiovascular risks were compared between a group-administered ULT and a control group reported that ULT had little effect in preventing cardiovascular events. ULT significantly reduced the total incidence of cardiovascular events and hypertension, but the risks of MACE and mortality were not improved [119,120]. The effect of ULT on atherosclerosis has been contradictory among studies. A large-scale RCT to investigate the effect of allopurinol on cardiovascular outcomes in patients with ischemic heart diseases was performed in the UK (ALL-HEART study). The results are yet to be published [121].

In addition to these urate-lowering agents, the efficacy of colchicine and IL-1β antagonist for the prevention of CV events was proven by large-scale RCTs. Canakinumab, an IL-1β monoclonal antibody, led to a significantly lower rate of recurrent cardiovascular events in patients with previous myocardial infarction and hsCRP levels of 2 mg or more per liter [88]. The administration of colchicine, an inflammasome inhibitor, also improved the prevention of CV events in patients with acute and chronic coronary diseases [89,122]. However, these drugs are not typically used for the long-term treatment of gout. If deposition of MSU is related to cardiovascular events, long-term use of colchicine may be considered for the prevention of gout attacks and cardiovascular events.

**Table 1 ijms-22-12394-t001:** Clinical trials for the efficacy of therapeutic agents of gout in cardiovascular risks.

Drugs	Study Design	Control	Participants	Number	Results
**Allopurinol**	RCT [107]	Usual therapy	Patientis with CKD (eGFR < 60 mL/min).	113	Allopurinol slows down the progression of renal disease and reduces risk of cardiovascular events by 71%.
	RCT [108]	Placebo	Adults with stage 3 or 4 CKD and no history of gout.	369	Allopurinol did not slow the decline in eGFR as compared with placebo.
	Meta-analysis: 12 RCTs [123]	Placebo or no treatment	RCTs investigated allopurinol’s effects on endothelial function. Patients with CHF, CKD, or type 2 DM.	CHF; 197 CKD; 183 DM; 170	Allopurinol had a benefit on endothelial function in patients with CHF and CKD but not in type 2 DM.
	Meta-analysis: 9 RCTs [109]	Placebo or control	Patients undergoing CABG, after ACS or CHF.	850	Allopurinol was associated with the reduction of odds of periprocedural ACS but not with that of long-term secondary prevention of ACS.
**Febuxostat**	RCT [116]	Allopurinol	Patients with gout and cardiovascular disease.	6190	All-cause and cardiovascular mortality were higher in the febuxostat group than in the allopurinol group (HR for all death, 1.22; HR for cardiovascular death, 1.34).
	RCT [117]	Allopurinol	Patients were ≥60 y.o., already receiving allopurinol, and had at least one additional cardiovascular risk factor.	6128	Febuxostat is non-inferior to allopurinol therapy as the primary cardiovascular endpoint and not associated with an increased risk of death.
**Xanthine oxidase inhibitor (XOI)**	Meta-analysis: 81 RCTs [119]	Placebo or no treatment	RCTs comparing purine-like or non-purine XOI with placebo or no treatment (control) for a period equal or superior to 28 days in adult patients.	10,684 (6434 pt·yr)	XOI did not significantly reduce the risk of MACE and death but reduced the risk of TCE and hypertension.
**Urate-lowering treatment (ULT)**	Meta-analysis: 18 RCTs [120]	Placebo or other ULT drugs	RCTs had to report cardiovascular safety of urate-lowering treatment (allopurinol, febuxostat, pegloticase, rasburicase, probenecid, benzbromarone).	7757	Any ULT did not demonstrate a significant difference in any cardiovascular death, non-fatal myocardial infarction or non-fatal stroke, or all-cause mortality.
**Colchcine**	RCT [122]	Placebo	Patients suffered from MI within 30 days.	4745	The primary endpoint occurred at 5.5% in the colchicine group compared with 7.1% in the placebo group (HR 0.77, 95% CI 0.61–0.96).
	RCT [89]	Placebo	Patients had any evidence of coronary disease and have been in a clinically stable condition for at least 6 months.	5522	A primary endpoint event occurred in 6.8% in the colchicine group and 9.6% in the placebo group (HR 0.69, 95% CI 0.57–0.83).

RCT, randomized controlled trial; CKD, chronic kidney disease; CHF, chronic heart failure; DM, diabetes mellitus; CABG, coronary artery bypass graft; ACS, acute coronary syndrome; MI, myocardial infarction; pt·yr, patient-years; HR, hazard ratio; CI, confidence interval; MACE, major adverse cardiovascular event; TCE, total cardiovascular event.

## Figures and Tables

**Figure 1 ijms-22-12394-f001:**
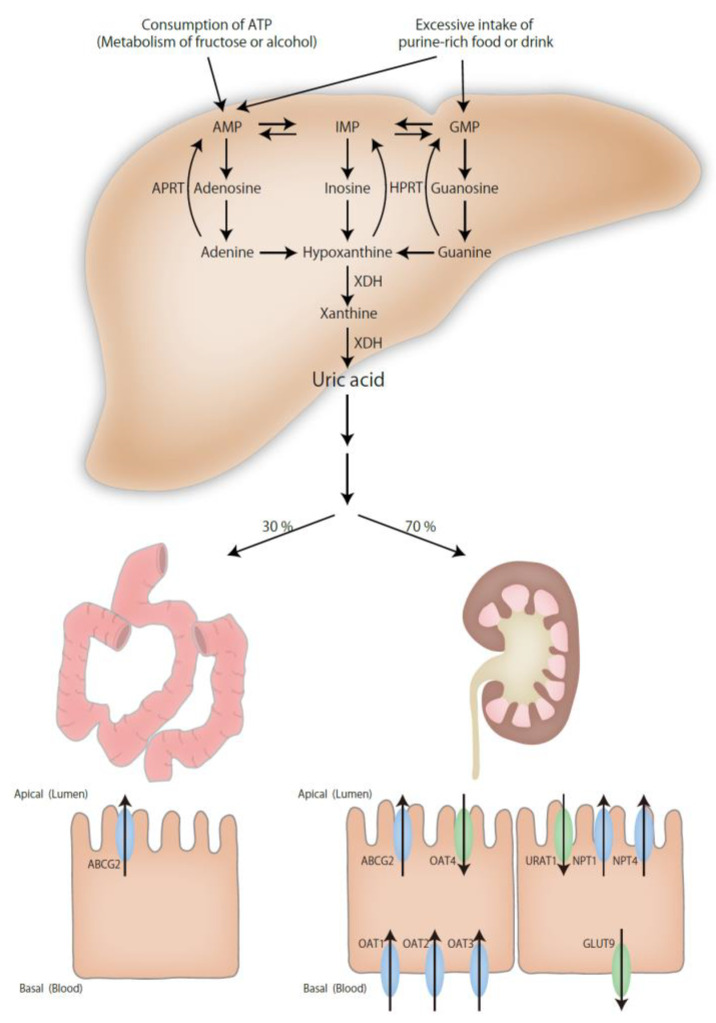
Mechanisms of hyperuricemia. Purine is metabolized to uric acid. Uric acid is excreted via the kidney and the intestinal tract. HPRT, hypoxanthine-guanine phosphoribosyl transferase; APRT, adenine phosphoribosyltransferase; XDH, xanthine dehydrogenase.

**Figure 2 ijms-22-12394-f002:**
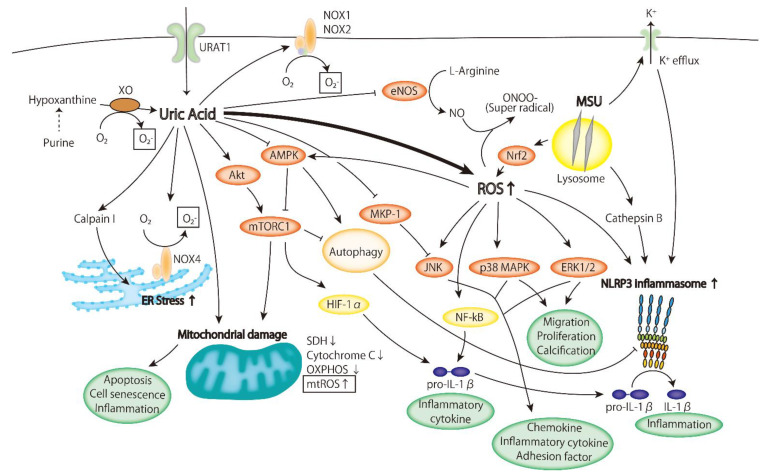
The effect of uric acid on intracellular signaling pathways in the pathogenesis of atherosclerosis. Intracellular uric acid induces reactive oxygen species (ROS) production and activates several inflammatory signaling pathways. XO, xanthine oxidase; NOX, NADPH oxidase; eNOS, endothelial NO synthase; MSU, monosodium urate; AMPK, AMP-activated kinase; Nrf2, Nuclear factor-erythroid 2-related factor 2; mTORC1, mammalian target of rapamycin complex 1; p38 MAPK, p38 mitogen-activated protein kinase; MKP-1, MAPK phosphatase-1; JNK, Jun N-terminal kinase; ERK, extracellular signaling-regulated kinase; HIF-1α, Hypoxia Inducible Factor 1α; SDH, succinate dehyderogenase; OXPHOS, oxidative phosphorylation; mtROS, mitochondrial ROS.

**Figure 3 ijms-22-12394-f003:**
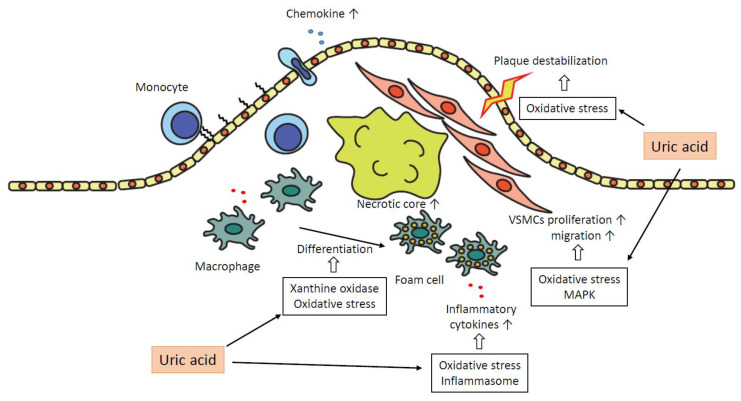
The role of uric acid in the formation of atheroma plaque. Uric acid plays a pro-atherogenic role in several steps in the progression of plaques. MAPK, mitogen-activated protein kinase; VSMCs, vascular smooth muscle cells.

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
