# Peer review of "Uric Acid in Inflammation and the Pathogenesis of Atherosclerosis"

_ijms, 2021, doi:10.3390/ijms222212394_

Round 1

Reviewer 1 Report

The review presented in the article is very complete and figures are nice illustrated and are very useful to understand all the mechanisms explained in the text. I think that the article is publishable in his current form and it is very interesting for the readers.

Author Response

Thank you for your great comment. It’s a great pleasure for us.

Reviewer 2 Report

This a comprehensive review. the current version is very confused and hard to be followed, which need to be organized. This review addresses the role of uric acid in the development of atherosclerosis. It is much beneficial to readers focusing on how uric acid plays an facilitating role in the foam cell formation, lipid core enlargement,  plaque stability, and clinical presentations. It is well-known that atherosclerosis is an inflammatory  complex event. the authors should elaborates the signaling role of uric acid in the different stages of this inflammatory disease based on the current tile for clarity in both basic and clinical studies.

Author Response

     We really appreciate your comment. We agree that it is beneficial for readers to have a review organized in a way how uric acid plays role in each process of the development of atherosclerosis. Nevertheless, we think that the notable point of our review is to summarize the roles of uric acid in each signaling pathway related to pathogenesis of atherosclerosis. Few past reviews have shown the role and relationships of intracellular signaling pathway related to uric acid clearly.

     As the reviewer suggested, the description of roles of uric acid in each signaling pathways helps the researchers involved in uric acid the further understanding of the relationship of uric acid with atherosclerosis and inflammation. Unfortunately, the evidence of the role of the uric acid in each process of the development of atherosclerosis is limited. Still, we looked into the literature and added a paragraph (page 11, Line 441-454.) and a summary figure (Figure 3) as for the role of uric acid in each stage of atherosclerosis.

Round 2

Reviewer 2 Report

I have no further comments.